# Improving Cuticle Thickness and Quality Traits in Table Grape cv. ‘Italia’ Using Pre-Harvest Treatments

**DOI:** 10.3390/plants13172400

**Published:** 2024-08-28

**Authors:** Paolo La Spada, Alberto Continella, Eva Dominguez, Antonio Heredia, Alessandra Gentile

**Affiliations:** 1Dipartimento Agricoltura Alimentazione e Ambiente (Di3A), Università degli Studi di Catania, 95123 Catania, Italy; 2Instituto de Hortofruticultura Subtropical y Mediterránea La Mayora, Universidad de Málaga—Consejo Superior de Investigaciones Científicas, Departamento de Mejora Genética y Biotecnología, Estación Experimental La Mayora, Algarrobo-Costa, E-29750 Málaga, Spain; 3Instituto de Hortofruticultura Subtropical y Mediterránea La Mayora, Universidad de Málaga—Consejo Superior de Investigaciones Científicas, Departamento de Biología Molecular y Bioquímica, Universidad de Málaga, E-29071 Málaga, Spain

**Keywords:** cracking, cuticle, calcium chloride, phytohormones, brown algae, table grape, *Ascophyllum nodosum*, salicylic acid

## Abstract

Table grape viticulture, due to the impact of climate change, will have to face many challenges in the coming decades, including resistance to pathogens and physiological disorders. Our attention was focused on fruit cracking due to its ubiquitous presence in several species. This study explores the effects of three different treatments on the epidermis and cuticle of table grape berries by evaluating the impact of the girdling technique on various fruit quality parameters, including cuticle thickness, sugar content, acidity, color, bunch weight, and rheological properties. The treatments were (1) calcium chloride (CaCl_2_), (2) calcium chloride + salicylic acid (CaCl_2_ + SA), and (3) calcium chloride + *Ascophyllum nodosum* (CaCl_2_ + AN), with and without girdling, plus an untreated control. This research was conducted over the 2021–2022 growing season in a commercial vineyard in Licodia Eubea, Sicily, Italy. The results indicate significant variations in cuticle thickness and other qualitative traits throughout the growth and ripening phases, with notable differences depending on the treatment used. This study’s findings suggest that specific treatments can influence the structural integrity of the grape cuticle, potentially impacting the fruit’s susceptibility to cracking and overall marketability. The findings provide valuable insights into the role of chemical treatments and cultural techniques in enhancing fruit quality and resistance to environmental stresses in table grape cultivation.

## 1. Introduction

Table grape represents a significant fruit crop globally and is a fundamental sector of the world economy. It has expanded markedly in recent decades due to various factors, including enhanced international trade, rising global incomes, advancements in technology, and heightened awareness of the health advantages associated with consuming fruits and vegetables [1]. Over the past decade, the world production of table grapes has amounted to 31.5 million tons. Presently, China is the primary producer, with India, Turkey, Egypt, and Iran contributing substantially to the total production [2]. Table grape viticulture will have to face numerous challenges, including resistance to pathogens (e.g., fungi, bacteria, viruses) and tolerance to physiological disorders, among which fruit cracking is a significant issue in vineyards. Fruit cracking can cause the fruit to look unappealing, diminish its quality and shelf life, and increase its vulnerability to biotic infections, ultimately resulting in fruits that cannot be sold [3]. Generally, fruit cracking is a common physiological disorder of horticultural crops, including sweet cherry, apple, litchi, pomegranate, citrus, plum, grape, fig, and tomato [4,5,6,7]. Aside from affecting the fruit’s appearance, cracking can also make the fruits more susceptible to biotic infection, thus causing significant economic losses [8]. It is forecasted that the frequency of extreme weather events will increase in the coming years, e.g., excessive rainfall [9]; these extreme events may increase the incidence of fruit cracking. Cracking susceptibility includes internal factors within the fruit itself and external environmental factors, such as cultivation management, irrigation, nutrition, and environmental conditions, e.g., temperature, wind, and light [4,5,10]. In addition, epidermal thickness, the sub-epidermis, and the cuticle have been reported to play a role in fruit cracking, and their interactions with environmental factors have also been studied [11,12,13,14,15].

A pivotal and critical role is played by the plant cuticle, which is an outer membrane that covers all non-woody aerial parts of the plants. It is an efficient barrier to water loss, attenuates ultraviolet light absorption, and protects the plant from biotic and abiotic environmental stresses [15]. The cuticle has a composite nature; it is composed of a lipid matrix named cutin intertwined with cell-wall polysaccharides and with embedded phenolics and waxes. The cuticle exhibits notable changes in the deposition, composition, and physicochemical properties among species, as well as during its growth and development [16,17].

If the cuticle cannot fully protect the plant from excessive water loss or uptake, pest attacks or environmental stresses could more efficiently affect yield, depreciating the fruits or even rendering them unmarketable. Cuticle mechanical properties are also important for fungal pathogen penetration by highly localized appressorial pressure [18], for oviposition substrate choice by insects [19], and for susceptibility to fruit cracking [20,21].

Several authors have dealt with mineral compounds, in particular, calcium in its various forms [22,23,24,25,26,27] and plant growth regulators [28,29,30,31,32]. In particular, gibelleric acid (GA) has been used in a lot of different species to reduce fruit cracking, as shown by some authors [33,34,35,36]. Regarding the application of GA_4+7_, the authors of [32] found that in mixture with 6-benxyladenine (BA), it may affect the peel characteristics of cv. ‘Pink Lady’ in reducing calyx-end cracking, while biostimulants [37,38,39,40,41,42,43] have been shown to improve cuticle features and resistance to cracking, increasing quality and resistance to abiotic and biotic stress both pre- and post-harvest. This study aimed to assess the effects of various treatments on the cuticle thickness, fruit quality (including the sugar content, acidity, and color), bunch weight, and rheological properties of the fruits. In detail, we investigated the effects on the cuticle of the ‘Italia’ table grape variety using three treatments with commercial products, namely (1) calcium chloride, CaCl_2_, (2) calcium chloride + salicylic acid girdled, CaCl_2_ + SA, and (3) calcium chloride + *A. nodosum*, CaCl_2_ + AN, using untreated plants as the control. Additionally, we have explored in each treatment the girdling technique and how it influenced the cuticle thickness, fruit quality (including sugar content, acidity, and color), bunch weight, and the rheological properties of the fruits.

## 2. Results and Discussion

In this work, the approach used was focused on the effects of the compounds and the girdling technique on the thickness of the cuticle and the changes in the fruit’s qualitative traits during the development and ripening phases of the cv. ‘Italia’ table grape. Our results are divided into two main categories for clarity. Section 2.1 examines the histological results, focusing on the thickness of the cuticle. Section 2.2 addresses the quality aspects, including chemical, rheological, and colorimetric parameters, to provide a comprehensive overview of the qualitative characteristics of the analyzed samples.

### 2.1. Histological Results

As illustrated in Figure 1, a difference in the cuticle thickness during the growth phases of the ‘Italia’ variety was observed in the untreated vines. Specifically, the cuticle thickness decreased from the third sampling (BBCH75), which exhibited the lowest values (2.19 µm), to the ripening phenological phase (BBCH89). The cuticle thickness varied during grape berry development, as reported by [44,45], indicating that the decline in thickness cuticle accompanies the fruit growth, thus resulting in one of the factors that affects fruit cracking. The daily growth accompanied by a decreasing cuticle thickness can favor microfractures [46], which, in the last stages of growth, can lead to fruit cracking [47] and, consequently, increase susceptibility to biotic and abiotic stress factors. Cuticle thickness can vary widely among grape cultivars, e.g., in ‘Merlot’ berries, it did not change significantly between the hard green and the onset of ripening stages, while it increased in ‘Concord’ berries [47]. The influence of the phenological stage on ‘Italia’ table grape cuticle thickness is further confirmed by a highly significant *p-*value (<0.0001).

The evaluation of the treatments started at the second phenological stage (BBCH73, peppercorn-sized berries). Figure 2 shows cross-sections of the fruit pericarp highlighting the cuticular layer differences at this stage for the different treatments, and Table 1 shows the differences in cuticle thickness. The grapes treated with CaCl_2_ + SA displayed the lowest cuticle thickness (4.32 µm), whereas the CaCl_2_ + AN treatment showed the highest (5.12 µm). However, no differences were observed between the other treatments (control and CaCl_2_) during this phase. A study conducted on apples using CaCl_2_ by [48] showed no change in the cuticle thickness compared to the control; this result may be related to our findings on CaCl_2_ in this phase. The results obtained in the ‘Italia’ table grape treated with CaCl_2_ + AN in this phenological phase seem to agree with the results reported by [37], where *A. nodosum* reduced the percentage of sweet cherry cracking damage in the ‘Skeena’ and ‘Sweetheart’ cultivars grafted onto Gisela 6. Also, the reduced cuticle thickness observed after the CaCl_2_ + SA treatment is similar to the results obtained by [49] in shallot, where the treatment with biosilica and salicylic acid had a negative impact on the cuticle thickness compared to the treatment with only biosilica. These results confirm the negative interaction between CaCl_2_ and salicylic acid found by [50] in various fruits.

Girdling is a technique widely used to obtain larger fruits [51,52]. The effect of girdling was studied in combination with the different treatments starting from the third phenological stage (BBCH75—pea-sized berry). Figure 3 shows images of pericarp fruit sections at this stage, highlighting the cuticular layer differences among the different combined treatments and the control. The cuticle thickness is presented in Table 2.

The vines treated with calcium chloride (CaCl_2_) showed no differences in cuticle thickness based on girdling. However, girdling did have an impact on the other treatments and the control. In these cases, girdling yielded a reduced cuticle thickness compared to the ungirdled vines. The results obtained in this phenological phase evince that the treatments with *A. nodosum* (CaCl_2_ + AN_U) and salicylic acid without girdling (CaCl_2_ + SA_U) resulted in a thicker cuticle than the CaCl_2_ + SA_G and CaCl_2_ + AN_G treatments. This denotes how *A. nodosum* may have a positive effect on the thickness of the cuticle, as reported by [37], which indicated a reduction in the cracking index of cherry. Salicylic acid, in contrast with the results in the previous phenological phase (BBCH73), had a positive effect on the cuticle thickness, as found by [21] on blueberry (*Vaccinium virgatum*), if not influenced by girdling. According to the results obtained in the previous phenological phase (BBCH73) and the results in [49], contextual factors, such as the different phenological phases, could play a role in modifying the plant’s response to the compounds and, consequently, the thickness of the cuticle. The girdling technique seemed to confirm its contribution to increasing susceptibility to cracking, as it has been reported by [5,53,54]. Interestingly, girdling did not seem to have an impact on the cuticle thickness in vines treated solely with calcium chloride, whereas in untreated vines, a negative effect was observed. This seems to point out a positive effect of calcium that counteracts the effect of girdling, which was also reported by [48] in apple.

During the phenological phase BBCH81 (beginning of ripening) (Figure 4 and Table 2), the effects of the different treatments were more notable (*p*-value < 0.0001). The lowest cuticle thickness was observed in the vines girdled and treated with calcium chloride, contrary to what was observed in the previous stage. This could be due to an acceleration in the ripening process caused by the girdling technique [55,56]. Again, unlike the previous stage, the cuticles from the girdled vines treated with CaCl_2_ showed a lower thickness (and similar to the girdled control) compared to those from the ungirdled vines. These results seem to be a late effect of the girdling, which influenced the cuticle thickness. However, in the vines treated with salicylic acid and *A. nodosum*, no differences in cuticle thickness were observed between the girdled and ungirdled vines.

In the last phenological stage studied, BBCH89 (berries ripe for harvest) (Figure 5 and Table 2), no differences were observed due to girdling, except for the untreated control, where girdling yielded fruits with a lower cuticle thickness. Nevertheless, and despite not being different, the girdled vines yielded a lower cuticle thickness in the different treatments compared to the ungirdled ones. These results obtained with girdling indicate that there was an influence on cuticle thickness, although this may vary with the phenological stage of development. Regardless of the treatment and girdling, the thickness of the cuticle decreased with fruit growth.

### 2.2. Fruit Quality Assays

The penetration test conducted on the fruits at harvest (phenological phase BBCH89) assessed the effect of the treatments (*p-*value < 0.0001). The effect of girdling on the table grapes modified various parameters, but its effect on fruit texture is unclear, as also reported by [57]. Yahuaca et al. [58] found a firmness reduction of 9.1% in ‘Red Malaga’ grape, but according to our data, the treatment with CaCl_2_ and the girdling technique influenced the fruit’s resistance to penetration (4.63 N for CaCl_2__G; 3.61 N for CaCl_2__U) (Table 3). All other treatments exhibited an opposite trend to that observed with CaCl_2_, and the resistance to penetration was higher in the ungirdled compared to the girdled, even if no variations were observed. It seems that the influence of the compounds used for this test may have also influenced the response of the plants to the girdling. Interestingly, the results obtained in the CaCl_2_ + SA_U (5.02 N) treatment showed the highest value, confirming what was found in previous research [59] on the positive effect of salicylic acid on fruit firmness. Generally, it can be concluded that the resistance to penetration appeared to be influenced by both the treatment and the girdling technique.

Texture profile analysis (TPA) was conducted on four parameters; the data are shown in Appendix A. The parameters used for the TPA were hardness (the highest peak force measured during the first compression), whose mean values ranged between 18.261 and 23.101; cohesiveness (the area underneath the second compression curve divided by the area underneath the first compression curve), whose mean values ranged between 0.191 and 0.700; springiness (a ratio of a product’s recovery to its original height), whose mean values ranged between 1.343 and 1.937; chewiness (hardness × cohesiveness × springiness), whose mean values ranged between 6.357 and 8.600. The results found in the literature seem to be controversial; some of the compounds used singularly, such as salicylic acid on apricots [60], showed an effect on the fruit textural properties. A CaCl_2_ treatment at the post-harvest stage had a strong effect on frozen mangoes using the dipping technique, as reported by [61]. The effect of *A. nodosum* did not seem to be strongly related to textural properties, but it could increase firmness, delaying the maturation process [62]. Finally, the effect of girdling is controversial because it appears to be strongly dependent on the variety and the timing of application [57]. Our results indicate that neither the treatment nor the girdling technique influenced the parameters.

Our results demonstrated that the combination of girdling and the tested compounds did not produce an effect on textual properties.

Analyses of the total soluble solids (TSS), whose mean values ranged between 16.54 and 17.10, and titratable acidity (percentage of tartaric acid), whose mean values ranged between 0.77 and 0.87, did not manifest any treatment differences, as shown in Appendix A.

The weights of 54 bunches evaluated at ripening indicate a trend toward weight increase due to girdling (Figure 6).

However, the treatment involving CaCl_2_ + SA_G combined with the girdling technique showed an effect (*p-*value < 0.0001), with the average weight of the bunches exceeding 1 kg (average of 1.19 kg), as shown in Figure 6. This result is supported by those of authors who highlighted the importance of salicylic acid to prevent weight loss [63,64,65].

The results obtained from the colorimetric analysis of the fruits (Table 4) indicate that the brightness (L) presented a higher value in the untreated and ungirdled control, in accordance with [51], while all the other treatments were not different. The hue angle was not influenced either by the treatment nor by girdling. The color intensity, measured by Chroma, presented very interesting data regarding the treatment with *A. nodosum* and the interaction with the girdling: the lowest value was found in the girdled and the highest value in the ungirdled, in line with the ability of *A. nodosum* to delay the maturation of table grapes [62]. In the analysis, we evaluated the ‘a’ component, which serves as an indicator of the green/red intensity, with the observed values ranging from −3.5 to −4.5. The lower the value, the more intense the green appears. In the salicylic acid treatments, there was a remarkable difference between the girdled and ungirdled plants, with the first ones exhibiting the most intense green (−4.5). In contrast, the ungirdled plants showed a lighter green (−3.8). Interestingly, the treatment with *A. nodosum* demonstrated the opposite trend, with the ungirdled plants recording a deeper green (−4.3), while those girdled, a paler green (−3.5). The ‘b’ component measures the color spectrum between blue and yellow, and all values leaned toward yellow. The most prominent yellow was observed in the treatment combining calcium and *A. nodosum*, which showed the highest value. The remaining treatments did not show significant differences. Overall, the differences may be linked to the secondary metabolism induced by the tested compounds [66].

## 3. Materials and Methods

### 3.1. Sampling

This study was conducted in a commercial vineyard in the countryside of Licodia Eubea (CT), Sicily, at GPS coordinates N 37°07′40.4″ and E 14°37′24.8″ during the 2021–2022 growing season. The vineyard, planted in 2015, was of cv ‘Italia’ table grape grafted onto RU 140 rootstock (Ruggeri 140). The vines were spaced 2.5 m apart within the rows and 2.5 m between the rows and trained using the Italian ‘Tendone’ overhead system [67]. The experimental design was a randomized block design with 54 vines in total, with 5 vines per treatment replicated 3 times each, as depicted in Figure 7. To mitigate edge effects, the field portions were centrally located, avoiding the field’s perimeter. The trial was conducted over the growing seasons from May to September 2021. Each replicate consisted of 15 vine trunks, excluding the 3 vines of a control with similar vegetative and productive characteristics. The treatments were applied using a battery-operated manual pump to ensure precision and minimize drift and included calcium chloride (CaCl_2_), calcium chloride combined with salicylic acid (CaCl_2_ + SA), and calcium chloride combined with *A. nodosum* (CaCl_2_ + AN). The control plants were treated only with water (C). After the second sampling, the girdling technique [52] was performed at the phenological phase BBCH73 (berries peppercorn size). Out of the 54 vines involved in this trial, we performed the girdling (G) technique on 26 of them (CaCl_2__G, CaCl_2_ + SA_G, CaCl_2_ + AN_G, and C_G) to assess its impact on the cuticle thickness of the fruits and development and quality parameters. The harvest was conducted on the same day for all the treatments. The method utilized to assess fruit ripeness was the value of the total soluble solids, which should be above 15, measured using a portable digital refractometer (Atago PR-32, Atago Ltd., Tokyo, Japan) [68].

The concentrations of chemical compounds were reported on the product labels for calcium chloride (300 g/hl) and *A. nodosum* (1.5 L/ha), while for salicylic acid, 1 mmol L^−1^ was applied as reported by [69]. We took samples during five phenological stages: BBCH71 (fruit set), BBCH73 (peppercorn-sized berries, bunches begin to hang), BBCH75 (pea-sized berries, bunches hang), BBCH81 (beginning of ripening, berries begin to develop variety-specific color), and BBCH89 (complete ripening) [70].

### 3.2. Chemical Compounds

To perform the field study, we used two commercial products: one for calcium chloride (7% NaCl_2_), and the other one for an *Ascophyllum nodosum* product consisting of 100% seaweed extract, which also contained 0.2% organic carbon and 0.7 g/L of mannitol. The salicylic acid used to prepare the solution was a high-purity laboratory-grade reagent (99% purity) known for its excellent solubility, produced by Sigma-Aldrich, USA.

### 3.3. Fruit Quality Analyses

Once ripe, a total of 54 bunches were collected and were singularly weighed. The total soluble solids (TSS, °Brix) and titratable acidity of tartaric acid (TA, expressed in %) of the fruit juice were measured by a Brix–acidity meter (model PAL-BX/ACID F5, Atago, Japan) on 3 independent replicates of 10 fruits for each treatment.

The color was determined using 30 fruits per treatment in triplicate with a Minolta colorimeter (CR-400; Minolta Camera Co., Tokyo, Japan). The wavelength used to perform this analysis was 400–750 nm. The fruits selected for the colorimetric analysis were placed transversely on the instrument, taking care to avoid any ambient light interference with the measurements, and ensuring that the fruit was in perfect contact with the device. Using the CIE (Commission International de l’Eclairage, Vienna, Austria) (L*, a*, b*) color scale [71], the Chroma (color saturation, Chroma = (a*2 + b*2)1/2) and hue angle (H° = tan − 1 (b*/a*)) were calculated.

### 3.4. Texture and TPA Analyses

The texture of 30 fruits in triplicate per treatment was measured using a texture analyzer (TA.XT. plus-Stable Micro System, Godalming, UK), and the data were analyzed with the instrument software “Exponent” (version 6,1,27,0). A load cell of 30 Kg and a cylindrical probe with a 2.0 mm diameter were used. The maximum force required to compress 5 mm of pulp at a speed of 1 mm s^−1^ was recorded using the protocol described by [72].

Another 30 fruits in triplicate per treatment were also used to conduct a TPA (texture profile analysis) on the parameters of hardness, cohesiveness, springiness, and chewiness using a compression plate and a probe with a diameter of 100 mm (P100).

### 3.5. Embedding Protocol

Five fruits were randomly selected from the bunch from each treatment in each phenological stage and embedded with the JB4 embedding kit protocol, as recommended by the producer (Polyscience Inc., USA). The selected samples were cut into small pieces (<2 mm) to enhance the penetration of the embedding resin and achieve a higher quality of the embedded samples.

### 3.6. Brightfield Microscopy and Image Analyses

The resin-embedded samples were cut into 3 µm-thick slices using a manual microtome (Leica, Wetzlar, Germany, RM2125). Each sample was triplicated, resulting in a minimum of 21 to 24 cut sections per slide. The sections were dried at 47 °C and then stained with Sudan IV dye (Sigma-Aldrich, USA) to highlight the cuticle. Subsequently, the samples were examined under a bright-field optical microscope (Leica, Wetzlar, Germany, DM 2500), and the cuticle thickness was determined using Fiji software version 2.9.0 [73]. A minimum of 50 measurements per sample, phenological stage, and treatment were analyzed.

### 3.7. Statistical Analysis

Statistical analyses were conducted utilizing Minitab software version 20. The general linear model (GLM) test was applied to all data, with the Bonferroni post hoc test employed to adjust for multiple comparisons, using a significance threshold of *p-*value < 0.05.

## 4. Conclusions

This research contributes to a better understanding of fruit cracking disorder by providing insights into the development of cuticle thickness in ‘Italia’ table grapes during growth and ripening. We observed a significant decrease in the cuticle thickness across five developmental stages, confirming that thinner cuticles can lead to fruit cracking and increased susceptibility to various stresses. The impacts of different treatments, especially those involving calcium chloride (CaCl_2_), salicylic acid (SA), and *A. nodosum* (AN), on cuticle thickness highlight their significant roles in preserving the structural integrity of the grape cuticle during critical growth phases. Notably, the combination of CaCl_2_ and AN generally resulted in a thicker cuticle, indicating the effect of *A. nodosum* in enhancing the fruit’s structural resilience and potential resistance to cracking. In contrast, the effects of salicylic acid were subtler, showing potential benefits especially when not combined with girdling, suggesting that treatment combinations can influence table grape’s resistance to various stresses. Our study also explored the effects of girdling, confirming its potential to enhance bunch weight, although the outcomes varied depending on the treatment applied and the timing of application. Beyond structural changes, our investigation extended to other quality parameters of the fruit. Key findings included improvements with treatments using CaCl_2_ + SA, the fruits of which exhibited the greatest firmness and an increase in bunch weight when combined with girdling. Although the treatments had a relatively mild impact on the fruit color, possibly due to changes in the secondary metabolism, the texture profile analysis (TPA) showed no significant differences among the treatments, indicating that the concentrations and methods used may not sufficiently alter the textural properties. Some of our findings offer significant insights for further investigation into the role of salicylic acid and the seaweed *A. nodosum* in enhancing the physical characteristics of the fruit cuticle to ensure structural stability. These should be examined independently of the interactions with calcium chloride and without girdling at various concentrations to assess their effects on the table grape cv ‘Italia’.

## Figures and Tables

**Figure 1 plants-13-02400-f001:**
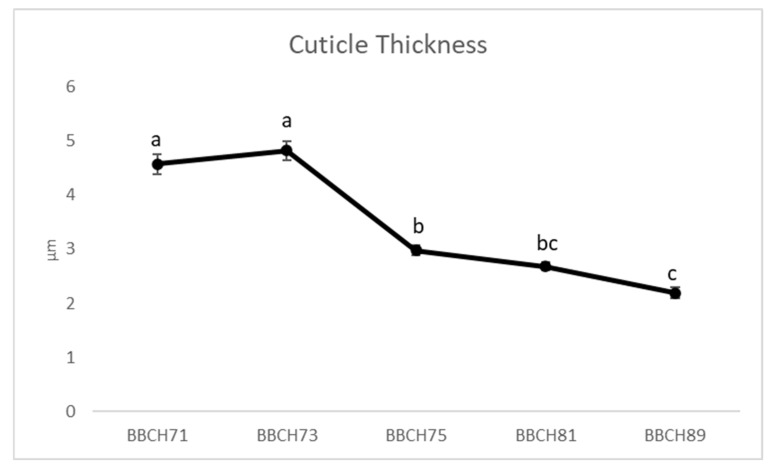
‘Italia’ fruit cuticle thickness (µm) among the phenological stages of BBCH71 (fruit set), BBCH73 (peppercorn-sized berries, bunches begin to hang), BBCH75 (pea-sized berries, bunches hang), BBCH81 (beginning of ripening) BBCH89 (complete ripening). Data presented as the means ± (SE). Different letters indicate significant differences (*p*-value < 0.05).

**Figure 2 plants-13-02400-f002:**
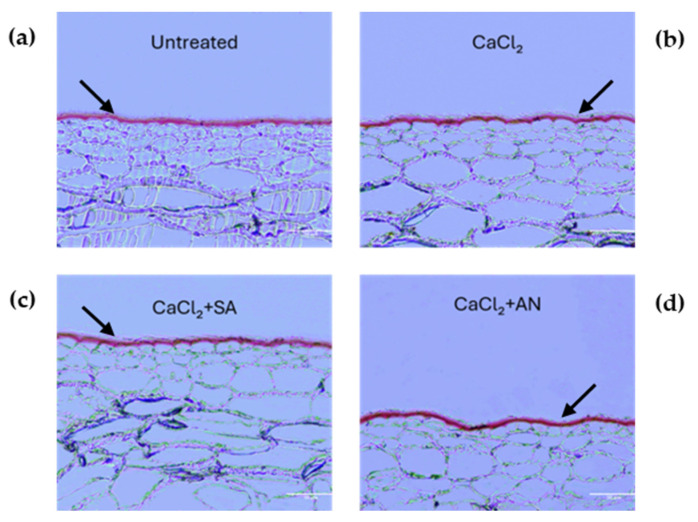
‘Italia’ table grape berries. Sections of 3 µm dyed red with Sudan IV to highlight the cuticle at the phenological phase BBCH73 (peppercorn-sized berries, bunches begin to hang). Magnification: 20X; scale bar: 50 µm. SA: salicylic acid; AN: *A. nodosum*. (**a**) Untreated; (**b**) calcium chloride; (**c**) calcium chloride + salicylic acid; (**d**) calcium chloride + *A. nodosum*. Arrows indicate the cuticles for better interpretation of the grape berry skin structure under different treatments.

**Figure 3 plants-13-02400-f003:**
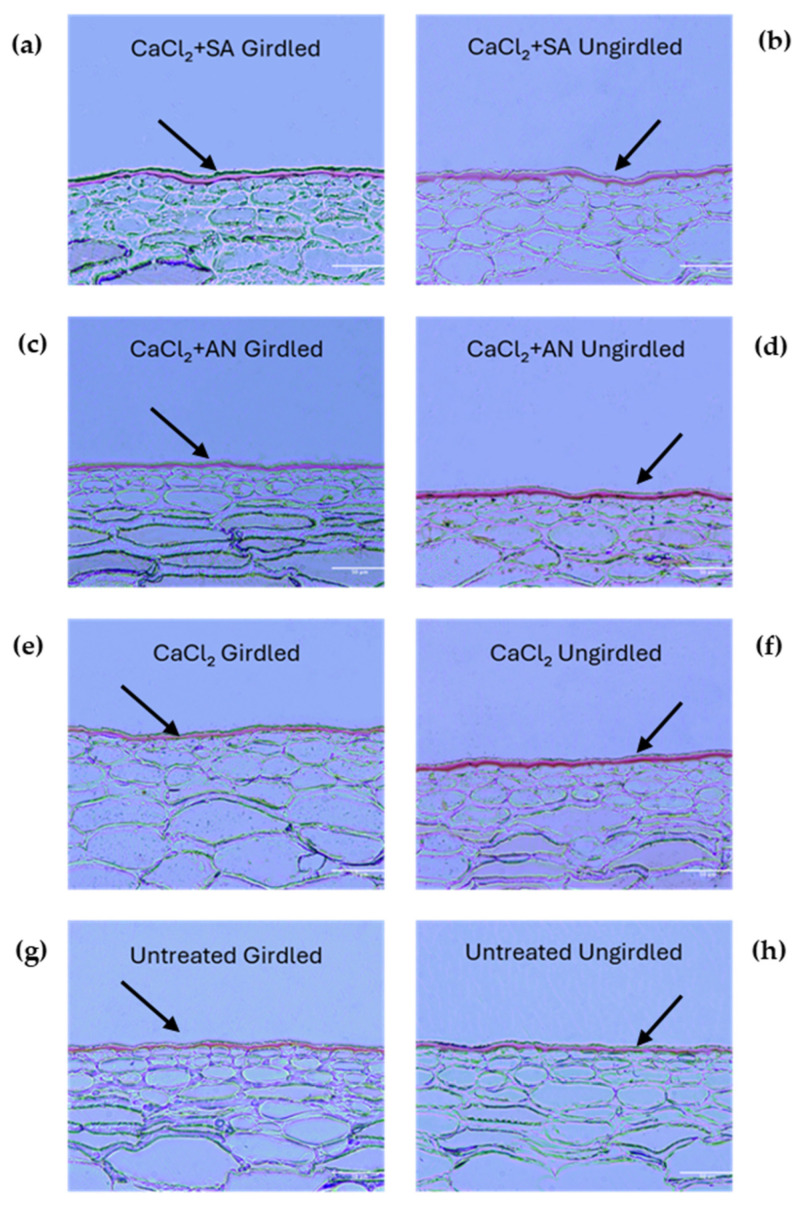
‘Italia’ table grape berries. Sections of 3 µm dyed red with Sudan IV to highlight cuticle at the phenological phase BBCH75 (pea-sized berries, bunches hang). Magnification: 20X; scale bar: 50 µm. SA: salicylic acid; AN: *A. nodosum*. (**a**) Calcium chloride + salicylic acid, girdled; (**b**) calcium chloride + salicylic acid, ungirdled; (**c**) calcium chloride + *A. nodosum*, girdled; (**d**) calcium chloride + *A. nodosum*, girdled; (**e**) calcium chloride, girdled; (**f**) calcium chloride, ungirdled; (**g**) untreated, girdled; (**h**) untreated, ungirdled. Arrows indicate the cuticles for better interpretation of the grape berry skin structure under different treatments.

**Figure 4 plants-13-02400-f004:**
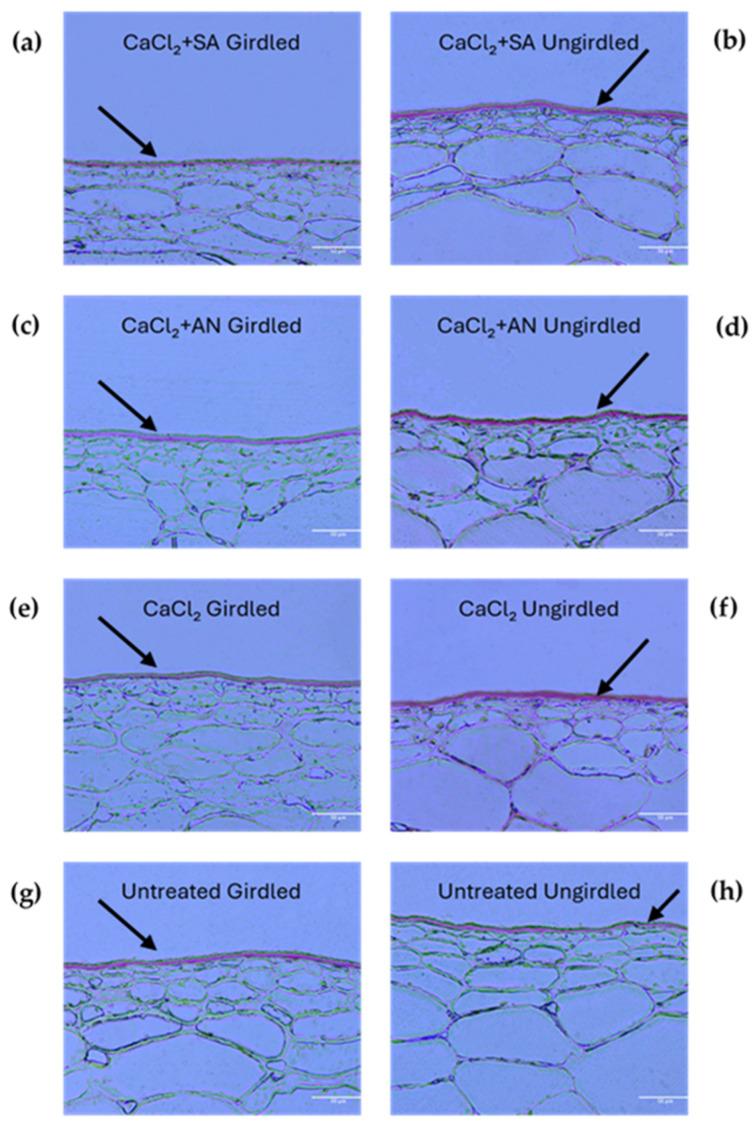
‘Italia’ table grape berries. Sections of 3 µm dyed red with Sudan IV to highlight cuticles at the phenological phase BBCH81 (beginning of ripening). Magnification: 20X scale bar: 50 µm. SA: salicylic acid; AN: *A. nodosum*. (**a**) Calcium chloride + salicylic acid, girdled; (**b**) calcium chloride + salicylic acid, ungirdled; (**c**) calcium chloride + *A. nodosum*, girdled; (**d**) calcium chloride + *A. nodosum*, girdled; (**e**) calcium chloride, girdled; (**f**) calcium chloride, ungirdled; (**g**) untreated, girdled; (**h**) untreated, ungirdled. Arrows indicate the cuticles for better interpretation of the grape berry skin structure under different treatments.

**Figure 5 plants-13-02400-f005:**
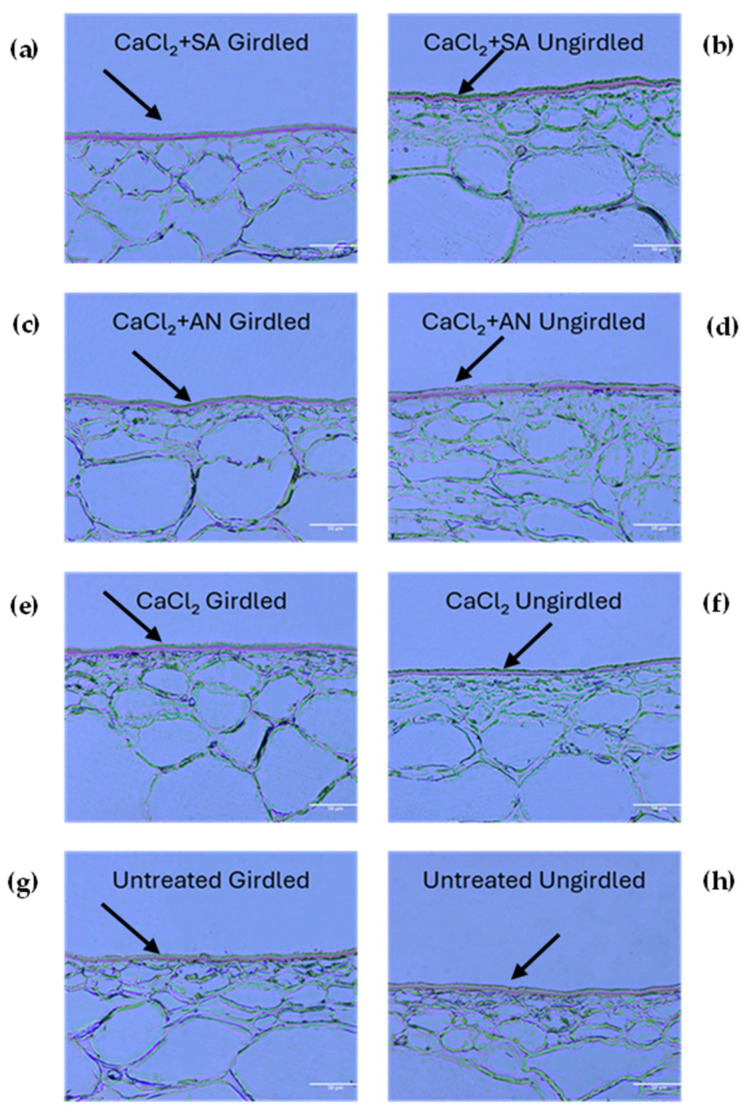
‘Italia’ table grape berries. Sections of 3 µm dyed red with Sudan IV to highlight cuticle at the phenological phase BBCH89 (complete ripening). Magnification: 20X; scale bar: 50 µm. SA: salicylic acid; AN: *A. nodosum*. (**a**) Calcium chloride + salicylic acid, girdled; (**b**) calcium chloride + salicylic acid, ungirdled; (**c**) calcium chloride + *A. nodosum*, girdled; (**d**) calcium chloride + *A. nodosum*, girdled; (**e**) calcium chloride, girdled; (**f**) calcium chloride, ungirdled; (**g**) untreated, girdled; (**h**) untreated, ungirdled. Arrows indicate the cuticles for better interpretation of the grape berry skin structure under different treatments.

**Figure 6 plants-13-02400-f006:**
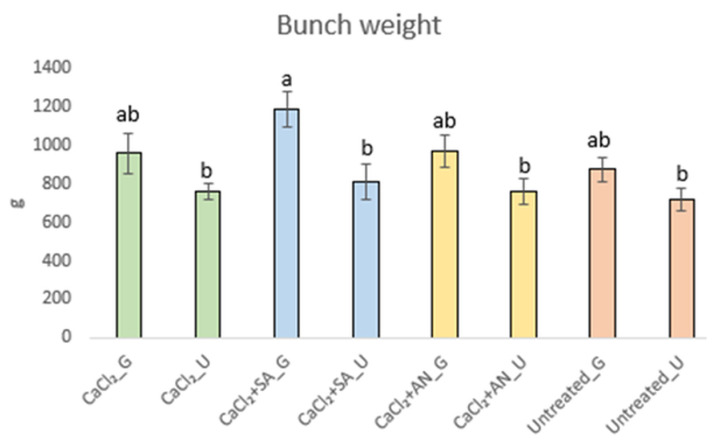
Bunch weights (g) of ‘Italia’ table grapes at the phenological stage BBCH89 (complete ripening). Data presented as the means ± (SE). Different letters indicate significant differences (*p*-value < 0.05). SA: salicylic acid; AN: *A. nodosum*; G: girdled; U: ungirdled.

**Figure 7 plants-13-02400-f007:**
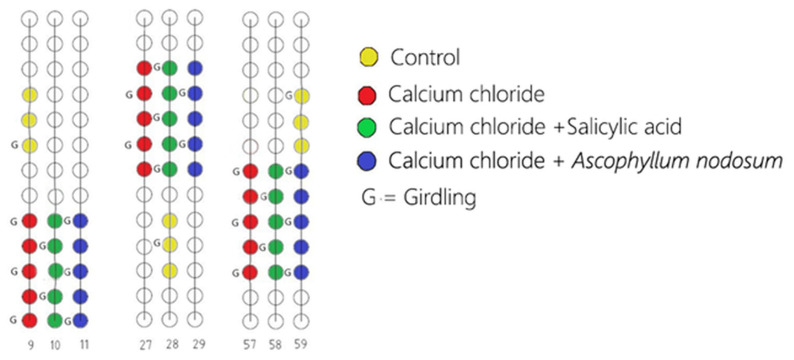
Scheme of the experimental design of the ‘Italia’ table grape field. CaCl_2_ (calcium chloride); CaCl_2_ + SA (calcium chloride + salicylic acid); CaCl_2_ + AN (calcium chloride + *Ascophyllum nodosum*); in both girdled (G) and ungirdled conditions.

**Table 1 plants-13-02400-t001:** Comparison of ‘Italia’ fruit cuticle thickness (µm) among different treatments at the phenological stage BBCH73 (peppercorn-sized berries, bunches begin to hang). Data presented as the means ± (SE). Different letters indicate significant differences (*p-*value < 0.05). SA: salicylic acid; AN: *A. nodosum*.

Treatment	Cuticle Thickness (µm) BBCH73
CaCl_2_	4.89 ± (0.22) ab
CaCl_2_ + SA	4.32 ± (0.11) b
CaCl_2_ + AN	5.12 ± (0.24) a
Untreated	4.82 ± (0.17) ab

**Table 2 plants-13-02400-t002:** Overview of ‘Italia’ fruit cuticle thickness (µm) data from three phenological stages: BBCH75 (pea-sized berries, bunches hang); BBCH81 (beginning of ripening); BBCH89 (complete ripening). Data presented as the means ± (SE). Different letters indicate significant differences (*p*-value < 0.05). SA: salicylic acid; AN: *A. nodosum*; G: girdled; U: ungirdled.

Treatment	Cuticle Thickness (µm) BBCH75	Cuticle Thickness (µm) BBCH81	Cuticle Thickness (µm) BBCH89
CaCl_2__G	3.31 ± (0.03) a	2.03 ± (0.05) e	1.71 ± (0.04) c
CaCl_2__U	3.32 ± (0.07) a	3.41 ± (0.08) a	1.93 ± (0.07) abc
CaCl_2_ + SA_G	2.81 ± (0.06) bc	2.53 ± (0.09) cd	2.05 ± (0.09) ab
CaCl_2_ + SA_U	3.40 ± (0.12) a	2.60 ± (0.04) cd	1.91 ± (0.06) abc
CaCl_2_ + AN_G	2.56 ± (0.06) bc	2.76 ± (0.08) bc	2.06 ± (0.07) ab
CaCl_2_ + AN_U	3.43 ± (0.10) a	2.93 ± (0.10) b	1.83 ± (0.042) bc
Untreated_G	2.49 ± (0.06) c	2.36 ± (0.06) de	1.76 ± (0.07) bc
Untreated_U	2.97 ± (0.09) ab	2.68 ± (0.07) bcd	2.19 ± (0.10) a

**Table 3 plants-13-02400-t003:** Penetration force data (N) of ‘Italia’ berries at the phenological stage BBCH89 (complete ripening). Data presented as the means ± (SE). Different letters indicate significant differences (*p*-value < 0.05). SA: salicylic acid; AN: *A. nodosum*; G: girdled; U: ungirdled.

Treatment	Penetration Force (N)
CaCl_2__G	4.63 ± (0.15) ab
CaCl_2__U	3.61 ± (0.12) d
CaCl_2_ + SA_G	4.66 ± (0.13) ab
CaCl_2_ + SA_U	5.01 ± (0.12) a
CaCl_2_ + AN_G	4.21 ± (0.15) bc
CaCl_2_ + AN_U	4.73 ± (0.16) ab
Untreated_G	3.99 ± (0.12) cd
Untreated_U	4.54 ± (0.16) abc

**Table 4 plants-13-02400-t004:** Colorimetric parameters of ‘Italia’ berries at the phenological stage BBCH89 (complete ripening). Data presented as the means ± (SE). Different letters indicate significant differences (*p*-value < 0.05). SA: salicylic acid; AN: *A. nodosum*; G: girdled; U: ungirdled.

Treatment	L	a	b	Chroma	Hue°
CaCl_2__G	40.40 ± (0.20) b	−4.21 ± (0.10) cd	11.39 ± (0.13) ab	12.20± (0.15) abc	110.23 ± (0.38) a
CaCl_2__U	39.84 ± (0.24) b	−3.99 ± (0.10) abc	10.63 ± (0.17) bc	11.27± (0.20) cd	108.30 ± (1.60) a
CaCl_2_ + SA_G	39.69 ± (0.20) b	−4.54 ± (0.12) d	11.33 ± (0.20) ab	12.25± (0.21) ab	110.51 ± (1.18) a
CaCl_2_ + SA_U	40.56 ± (0.18) b	−3.86 ± (0.11) abc	11.36 ± (0.23) ab	12.02± (0.25) abcd	108.86 ± (0.44) a
CaCl_2_ + AN_G	39.72 ± (0.17) b	−3.51 ± (0.14) a	10.43 ± (0.17) c	11.07 ± (0.19) d	108.89 ± (0.47) a
CaCl_2_ + AN_U	40.28 ± (0.17) b	−4.34 ± (0.16) cd	11.95 ± (0.22) a	12.80 ± (0.25) a	110.22 ± (0.36) a
Untreated_G	40.20 ± (0.20) b	−3.65 ± (0.10) ab	11.05 ± (0.21) bc	11.68± (0.21) bcd	108.58 ± (0.46) a
Untreated_U	41.50 ± (0.23) a	−4.19± (0.12) bcd	11.42 ± (0.22) ab	12.19± (0.24) abc	108.93 ± (1.16) a

## Data Availability

The original data presented in this study are available at https://doi.org/10.17632/bt4yxs7w5k.1 (accessed on 15 July 2024).

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
