# Peer review of "Improving Cuticle Thickness and Quality Traits in Table Grape cv. ‘Italia’ Using Pre-Harvest Treatments"

_plants, 2024, doi:10.3390/plants13172400_

Round 1

Reviewer 1 Report

Comments and Suggestions for Authors

Dear Authors, I have read the MS entitled Improving cuticle thickness and quality traits in table grape cv. ‘Italia’ using pre-harvest treatments.

This study explores some effects of different treatments on the epidermis and cuticle of table grapes including cuticle thickness, sugar content, acidity, color, bunch weight, and rheological properties. The common "girdling" technique was also an experimental factor.

The study is of interest to table grape producers, as skin faults lead to losses in commercialisation.

I do have some observations, however,

-please check journal requirements regarding structure of MS, as material and methods is preffered to be before results and discussions usually

-I suggest adding a new subchapter regarding chemical and equipment used for the analysis, mentioning their purity and origin and any other details necessary

-please give a bit more details regarding analysis methods (for example, in the case of colour analysis, what wavelength was used, what vials were employed etc). It is not enough to just mention a reference, please give some details.

-should it be weight or mass? please check all over text

-please use trunks (vine trunks) instead of trees

-please clarify the measuring unit in TA (g/L what acid?). Also, table 5 shows questionable values for TA (g/L). Please check.

-

Comments on the Quality of English Language

Dear Authors,

English language is generally fine in the MS. There are a couple of small issues that need to be resolved:

-please do not use active voice: "We focused..." (abstract), We investigaed (line 75) etc. In scientific articles, passive voice is preffered.

-please tend to minor editing issues, like too many spaces or adding commas.

Reviewer 2 Report

Comments and Suggestions for Authors

The reviewed manuscript, entitled “Improving cuticle thickness and quality traits in table grape cv. ‘Italia’ using pre-harvest treatments” (plants-3156156), discusses an interesting aspect related to commercial grape cultivation in the context of climate change.

The manuscript is generally well-written, but several aspects require correction before it can be accepted for publication. I believe these corrections will certainly enhance the final quality of the publication.

1.     The methodology should be supplemented with information regarding the age of the bushes in the vineyard, the method used to determine grape ripeness for harvesting, and whether all combinations were harvested simultaneously (on the same day). Could the treatment have influenced the ripening date? Harvesting on different dates might have affected the results' quality, regardless of the experimental factors.

2.     Figures 1-5 (skin cross-sections) require a more detailed description and graphical indications of key features (e.g., with arrows) because, without these, they are difficult to read and interpret. The authors do not adequately address this in the text, apart from merely referring to these figures.

3.     In the Introduction, briefly discussing the use of GA4+7 as an agent to prevent skin cracking in some species, such as sweet cherries, would be beneficial.

4. Editorial issues require improvement: commas in numerical data should be replaced with full stops, and the spelling of "p-value" should be corrected (see notes in the attached PDF).

Reviewer 3 Report

Comments and Suggestions for Authors

I have attached a marked manuscript as well as a detailed review of this manuscript.  As you will see, I am recommending review with possibility of acceptance after re-submission and re-review.

Comments on the Quality of English Language

The English is OK but will require some text editing.  I would wait until a revised manuscript is submitted.  Perhaps authors should run it through someone whose first language is English.

Round 2

Reviewer 3 Report

Comments and Suggestions for Authors

Authors have addressed all my concerns with the original manuscript.

Comments on the Quality of English Language

English is adequate.